# Fecal Pollution Drives Antibiotic Resistance and Class 1 Integron Abundance in Aquatic Environments of the Bolivian Andes Impacted by Mining and Wastewater

**DOI:** 10.3390/microorganisms8081122

**Published:** 2020-07-26

**Authors:** Jorge Agramont, Sergio Gutiérrez-Cortez, Enrique Joffré, Åsa Sjöling, Carla Calderon Toledo

**Affiliations:** 1Environmental Microbiology Unit, Institute of Molecular Biology and Biotechnology, Universidad Mayor de San Andrés, La Paz, Bolivia; cvcalderon@fcpn.edu.bo; 2Department of Microbiology, Tumor and Cell Biology, Karolinska Institutet, 23109 Stockholm, Sweden; enrique.joffre@ki.se (E.J.); asa.sjoling@ki.se (Å.S.); 3Centre for Translational Microbiome Research, Karolinska Institutet, 23109 Stockholm, Sweden

**Keywords:** antibiotic resistance genes (ARGs, crAssphage, wastewater, fecal pollution, metal contamination

## Abstract

An increased abundance of antibiotic resistance genes (ARGs) in aquatic environments has been linked to environmental pollution. Mining polluted sites with high concentration of metals could favor the in situ coselection of ARGs, whereas wastewater discharges release fecal antibiotic resistant bacteria in the environment. To study the effect of human fecal contamination and mining pollution, water and sediment samples affected by mining activities and sewage discharges were collected from three lakes in Bolivia, the pristine Andean lake Pata Khota, the Milluni Chico lake directly impacted by acid mine drainage, and the Uru-Uru lake located close to Oruro city and highly polluted by mining activities and human wastewater discharges. Physicochemical parameters, including metal composition, were analyzed in water and sediment samples. ARGs were screened for and verified by quantitative polymerase chain reaction (PCR) together with the mobile element class 1 integron (*intl1*), as well as crAssphage, a marker of human fecal pollution. The gene *intl1* was positively correlated with *sul1*, *sul2*, *tetA*, and *bla_OXA-2_*. CrAssphage was only detected in the Uru-Uru lake, and its tributaries and significantly higher abundance of ARGs were found in these sites. Multivariate analysis showed that crAssphage abundance, electrical conductivity, and pH were positively correlated with higher levels of *intl1* and ARGs. Taken together, our results suggest that fecal pollution is the major driver of higher levels of ARGs and *intl1* in environments contaminated by wastewater and mining activities.

## 1. Introduction

Antibiotic resistance is considered a major threat to human health worldwide [1]. Antibiotic resistance genes (ARGs) have been classified as emergent contaminants, with a significant impact in aquatic environments due to the possibility to be acquired by pathogens, which could lead to public health issues [2]. Novel rearrangements of ARGs and mobile genetic elements (MGEs), that favor their dissemination, are considered xenogenetic pollutants. These elements can be incorporated and replicated in environmental microorganisms, thereby increasing their concentration [3].

It has been reported that anthropogenic activities cause pollution of aquatic environments with ARGs and MGEs [4]. Wastewater discharges cause the co-occurrence of MGEs and different ARGs in water and sediments [5]. At a continental scale, ARGs in sediments are strongly correlated with MGEs and antibiotic residues [6]. Recently, it was observed that microorganisms living in aquatic microbial communities that came from wastewater were able to transfer ARGs via horizontal gene transfer (HGT) after exposure to low levels of antibiotics and biocides [7]. Many of the ARGs that can be found in clinical settings have also been found in the environment, suggesting the possibility of movement and dissemination between these two scenarios [8].

Mining activities cause contamination of downstream water with dissolved metals [9], where heavy metals tend to accumulate into sediments [10]. Several studies have suggested that heavy metals can exert selective pressure to favor the coselection of ARGs, i.e., its simultaneous acquisition with metal resistance genes [11]. For example, urban soil samples of Belfast in Northern Ireland exhibited co-occurrence between metals (Zn, Cu, Cd, Co, Ni, Hg, Cr, and As) and many ARGs. Moreover, the degree of metal toxicity was positively correlated with the abundance of MGEs and ARGs [12]. Metals, such as Cu and Zn, can exert selection pressure over soil microbial communities to select resistant bacteria, even more than specific antibiotics [13]. In the Dongying river in China, Cu and Cr levels were positively correlated with the abundance of different ARGs [14], whereas both Zn and Pb levels were correlated with the abundance of erythromycin resistance genes in wastewater treatment plants [15]. These data suggest that aquatic environments are important for the ecology and evolution of ARGs. In particular, water bodies can be hotspots for the evolution of ARGs due to the convergence of antibiotics, microorganisms from different sources, biocides, and heavy metals [16], generating a scenario that favors the emergence, persistence, and dissemination of ARGs [17].

Levels of fecal pollution are not frequently considered in the analysis of the selection and dissemination of ARGs [18]. Thus, the incorporation of a molecular marker of human fecal pollution can help us to disentangle the accumulation of ARGs due to fecal bacterial discharges, and the ARGs selection and dissemination caused by other environmental contaminants [18]. CrAssphage most probably infect *Bacteroides* and *Prevotella* bacteria in the human gut [19]. The gene KP06_gp31 that belongs to crAssphage is highly abundant in aquatic environments contaminated by human feces, while it is less abundant in aquatic environments polluted by feces of other animals [20]. Thus, the crAssphage can be considered a marker for human fecal pollution.

For centuries, mining has been one of the most important economic activities in the Bolivian Andean region, severely impacting water resources [21,22]. This region also faces water scarcity due to the adverse effects of climate change on glaciers [23]. Besides, urban wastewater is directly released into the environment, polluting the water with enteric pathogens, antibiotics, and antibiotic-resistant bacteria [24,25,26]. Very high levels of antibiotics cause an evident selection in the environment for ARGs in sewage and sewage polluted sites [18]. However, to our knowledge, there are no previous studies about ARGs abundance in environments with very high metal levels and wastewater discharges.

This work aimed to analyze the relation of metal pollution and human fecal discharges on the abundance of different ARGs and the class 1 Integron (*intl1*), in water and sediments samples from pristine, mining-polluted, and wastewater mining-contaminated lakes, in order to explore the contributions of metals and fecal discharges in the abundance of ARGs. The class 1 integron is the most abundant and widely distributed integron among bacteria [27] and it is considered an anthropogenic pollution gene marker, commonly linked with both antibiotic and metal resistance genes in gram-negative bacteria [28]. Physicochemical parameters and metal levels were measured to account for mining pollution. Metal analysis included those elements previously reported in the study area [29,30] known for their coselection effects on antibiotic resistance [15,31,32,33,34,35].

## 2. Materials and Methods

### 2.1. Sampling Sites

#### 2.1.1. The Milluni Valley Lakes

The Milluni Valley is located in the Andean region of Bolivia, in the Department of La Paz, 20 km from La Paz city in the Cordillera Real. It is a glacial valley at the foot of the Huayna Potosi mountain. The valley has four lakes: Pata Khota (4670 masl), Jankho Khota (4575 masl), Milluni Chico (4540 masl), and Milluni Grande (4530 masl), the biggest one, with a surface of 2.37 km^2^ and depth of 4 m. Milluni Grande has a dam that captures and supplies water to the Puchucollo drinking water treatment plant, which is then distributed to the cities of La Paz and El Alto.

Mining activities were performed in the valley until 1990 by the company COMSUR, and water from the surrounding lakes was used in the mining activities. Acid mining drainage (AMD) was discharged directly in the Milluni Chico Lake, also contaminating the downstream Milluni Grande Lake. Consequently, these two lakes acquired an extremely acid pH that favored the mobility of metals (Cd, Zn, As, Cu, Ni, Pb, Sn) in water and sediments [29]. In contrast, the first lake, Pata Khota, is fed by water proceeding from the melting of Huayna Potosi mountain. Anthropogenic activities are minimal around this site, and the lake is considered an ecologically intact environment [29,36].

#### 2.1.2. Uru Uru Lake

The Uru Uru Lake (3686 masl), situated in the department of Oruro, in the central part of the Altiplano in Bolivia, is an artificial shallow lake 8 km south of Oruro city. The lake is characterized by an alkaline pH (8.3 ± 0.6) with a strong buffering capacity [30]. The Tagarete channel receives and drains untreated wastewater from Oruro city toward the northern part of the lake. The northeast part of Uru Uru Lake receives water discharges from San Jose Mine and the Vinto smelting plant [30]. On the other hand, the Desaguadero River that comes from the Titicaca Lake drains the discharges of Kori Kollo and Kori Chaca meromictic lakes (once open pit gold mines) into the northwest part of Uru Uru [37]. Previous studies in Uru Uru reported that the contribution of both wastewater and mining residues increased the electrical conductivity (EC) and the concentration of certain metals and metalloids such as Hg, Fe, Mn, W, and Sb [30].

#### 2.1.3. Sample Collection and Processing

Milluni samples were collected during the dry season in July 2016. Three points were randomly selected in the Milluni Chico (MC) and Pata Khota (PK) lakes (Figure 1). Electrical conductivity (EC) and pH were measured directly on the water (Oakton Instruments, Vernon Hills). Duplicate superficial sediment samples were collected in sterile 50-mL centrifuge tubes for both DNA extraction and metal quantification. Samples were immediately labeled and stored at 4 °C with cold packs and rapidly transported to the laboratory where they were stored at −70 °C until their analysis.

Samples of Uru Uru and its tributaries were collected at the beginning of the rainy season (November 2018). Three different points were considered: (1) UP1: The channel that discharges the water of the meromictic lake Kori Chaca into the northwest part of Uru Uru Lake. Agricultural activities are performed around this channel, and wastewater discharges were previously reported [38]; (2) UP2: The Tagarete channel that carries untreated wastewater discharges from Oruro city; and (3) UP3: Located in the northeast part of the lake, where Tagarete’s discharges drain.

Sediment samples were collected in triplicate. Parameters were recorded as described for Milluni. Surface sediment samples were collected in sterile 50-mL centrifuge tubes using a Core Sampling Device. The samples were divided into two fractions, one for DNA extraction and the other for metal quantification. Samples were kept at 4 °C with cold packs and transported to the laboratory in La Paz city, where they were rapidly stored at −70 °C until their analysis. Surface water samples from UP1, UP2, and UP3 were collected in triplicate and filtered (300 mL) through 0.45-µm nitrocellulose filter membranes (Sigma-Aldrich). The filters were immediately stored at −70 °C until their analysis.

#### 2.1.4. Quantification of Metals

Six elements were quantified in the sediment and acidified water samples: Cu, Zn, Pb, Ni, Cd, and As. All the analyses were performed as previously described [38]. The measurement was performed using inductively coupled plasma mass spectrometry. The quantification was performed at the Laboratorio de Calidad Ambiental (LCA), Universidad Mayor de San Andres.

#### 2.1.5. DNA Extraction

DNA was extracted from sediments using the PowerSoil DNA isolation kit (Qiagen, Germany). A prewashing step was performed using solution S0 (0.1 M EDTA, 0.1 M Tris (pH 8.0), 1.5 M NaCl, 0.1 M NaH_2_PO_4_, and Na_2_HPO_4_) [39], due to the acidity of some samples and the presence of heavy metals. Briefly, 300 mg of sediments were washed with 1.5 mL of solution S0 overnight in a horizontal shaker at 180 rpm at 4 °C, and the sediment was recovered by centrifugation at 12,000× *g* for 5 min and repeatedly washed with S0 until the supernatant appeared clear. After washing, the sediments were transferred to PowerBead Tubes (Qiagen, Germany), and the extraction proceeded as described by the manufacturer’s instructions.

Additionally, a quarter of the filtered water samples was used for DNA extraction using the PowerSoil DNA isolation kit (Qiagen, Germany). The filter was transferred into the *PowerBead* Tubes (Qiagen, Germany) to proceed with the DNA extraction according to the manufacturer’s protocol. DNA concentration was measured using Qubit^®^ dsDNA HS (Invitrogen, OR, USA).

#### 2.1.6. Quantitative PCR

The selection of ARGs for the analysis was performed as follows. The Antibiotic Resistance Genes Microbial DNA quantitative polymerase chain reaction (qPCR) arrays (Qiagen, Valencia, CA, USA) were used to screen for the presence of ARGs. The array consisted of 96 well-plates with predispensed primers for 85 different ARGs (Appendix A (Appendix A)) conferring resistance to antibiotics commonly used in clinical settings. Twelve positive ARGs (CT < 39) were selected to perform the assays of absolute quantification. Additionally, sulfonamides resistance genes (*Sul1* and *Sul2*) were included in the analysis as previous reports point at their presence in Milluni [26].

Standard curves for the absolute quantification of target genes were constructed using a plasmid as a template (Appendix A (Appendix A)). This plasmid was engineered by the insertion of the PCR assembled products of 14 ARGs (β-lactams (*acc-3*, *bla*_IMP-2_, *bla*_IMP-5_, *bla*_IMP-12_, and *bla*_OXA-2_), macrolide-lincosamide-streptogramin B (*msrA*), methicillin (*mecA*), quinolones (*qnrB1*, *qnrB5*, and *qnrS1*), tetracycline (*tetA* and *tetB*), and sulfonamides (*sul1* and *sul2*)), *intl1*, and the KP06_gp31 gene of the crAssphage, into the *Xba*I restriction site at the MCS of the pUC57 vector. The assembled sequence was synthesized and inserted by GenScript (GenScript, Piscataway, NJ, USA). Reference sequences for the ARGs were obtained from The Comprehensive Antibiotic Resistance Database (CARD) [40] and primers (Table 1) were designed using Primer-BLAST (NCBI) [41]. A six-point calibration curve was generated using serial dilution from 10^6^ to 10^1^ copies of the plasmid. The 16S rRNA housekeeping gene was used for the normalization of the absolute quantification of ARGs.

For qPCR experiments, the reaction mix consisted of 12.5 μL of Power SYBR^®^ Green PCR Master Mix (Applied Biosystems, Foster City, CA, USA), BSA 0.1 μg/μL (New England Biolabs, Ipswich, MA, USA), 10 pmol of each primer, and 2 μL of DNA, in a final volume of 25 μL. All qPCR experiments were performed in a LightCycler^®^ 480 Instrument II (Roche Molecular Diagnosis), with the following conditions: An initial denaturalization cycle at 95 °C for 5 min, followed by 45 cycles of denaturalization at 95 °C for 15 s, and amplification at 60 °C for 30 s. A melting curve was performed. All the qPCR experiments were performed at the Centre for Translational Microbiome Research (CTMR) Karolinska Institutet.

The absolute abundance of ARGs and *i**ntl1* gene was normalized to the absolute abundance of the 16S rRNA gene, as described before [6]. The normalized abundance was corrected in order to express the abundance of genes per bacteria, assuming that the average number of 16S rRNA genes per bacteria was four [6]. The absolute abundance of the KP06_gp31 gene (crAssphage) was considered for all the analyses.

#### 2.1.7. Statistical Analysis

All statistical analyses were performed in the Software R 3.6.1. (R Core Team) An analysis of variance (ANOVA) test was performed for the account of statistically significant differences in physicochemical parameters between sites. Previously, the distribution of residues and the homogeneity of variance were analyzed using quantile-quantile plot (*q-qPlot*) and diagrams of dispersion, respectively. When the data did not fit with the residue distribution and/or the homogeneity of variance, the ‘*Box-cox*’ function was used to choose an appropriated transformation. When the ANOVA test was significant, pairwise comparisons among means via post-hoc Tukey was performed to find out differences among sample sites. The package ‘*multcomp*’ (Version 1.4–10) was used for multiple comparisons. Pearson correlations were performed in order to evaluate the correlation among physicochemical parameters and quantified genes.

A heat map with hierarchical clustering ordination was performed to present the normalized abundance of the quantified genes, using the R package gplots (Version 3.0.1.1).

A Principal Component Analysis (PCA) was performed in order to examine the relationship between the quantified genes, the contributions of fecal discharges, the physicochemical variables, and the measured elements. Sampling points were ordered in function of the normalized abundance of ARGs and *intl1*. The two axes that explained most of the variation were extracted, and a multiparametric linear regression (Lm) was used to relate the ordination of the sampling points along the axis with the environmental variables. The selection of the best regression model was automatically performed using the function ‘*regsubsets*’ (backward and forward) from the package ‘*leaps*’ (Version 3.1) in function of R^2^ adjusted values. The package ‘*car*’ (Version 3.0–3) was used to calculate the variance inflation factor (VIF) of the independent variables to avoid multicollinearity. The packages’ *factoextra*’ (Version 1.0.6), and ‘*vegan*’ (Version 2.5–5) were employed for ordination analysis. Both datasets, the abundance of ARGs, *intl1*, and the explaining variables (EC, pH, metal levels, and the fecal pollution marker crAssphage) were logarithmically transformed before the analysis (S1 data (Appendix A)).

## 3. Results

### 3.1. Physicochemical Conditions and Metal Levels

We first evaluated the physicochemical characteristics and metal concentration on every sampling site. The measurement of EC and pH from water samples was performed in situ and before the collection of sediments. As shown in Figure 2a, the shallow water lake of Milluni Chico (MC) presented the lowest pH levels (2.32 ± 0.06 in 3 samples). The MC Lake showed AMD discharges and an intense orange color. The Pata Khota (PK) sampling sites and Uru Uru Lake had pH values close to neutral (Figure 2A). The EC measurements showed that PK samples registered the lowest EC values, while MC and Uru Uru (UP1, UP2, and UP3) water samples showed a significant 10- to 20-fold increase compared to PK, except for UP1 (Figure 2B).

Sediments were quantified for the presence of six elements (As, Cd, Pb, Ni, Cu, and Zn). The results (Figure 2C) indicated clear differences between sample sites in their metal composition. UP3 (Uru Uru Lake) had the highest concentration of Zn (1811 mg Kg^−1^ of sediment). In comparison with all other samples, statistically significant higher levels of As, Cd, Pb, Ni, and Cu were found in UP3 except for Cu levels in MC sediments. After UP3, MC samples were the second most abundant for all the elements analyzed, with significantly higher concentrations than UP2 and significantly higher concentrations of As, Cd, Ni, Cu, and Zn in relation to PK samples, and also significantly higher than UP1 except for As, Cd, and Pb levels. UP1 presented higher levels of As, Pb, Ni, and Cu than UP2. Thus, PK samples and UP2 were the sites with the lowest levels of all elements, except for Cu, with significantly higher concentrations in UP2. In summary, all the points directly impacted by mining activities (UP3 followed by MC sites and UP1) contained higher concentrations of As, Cd, Cu, and Zn.

A PCA was performed to visualize the distribution of the sampling points based on their environmental variables (Figure 2D). All the PK sampling points presented very similar values and were grouped together. A similar trend was observed for MC sampling points. In contrast, Uru Uru sampling points were differentiated by their physicochemical parameters and metal concentrations.

The correlation analysis of all the physicochemical parameters and metal levels (Figure 3) showed that metals levels were positively correlated among them. EC and pH showed a negative correlation between them, and no significant correlation with other environmental parameters was observed.

### 3.2. Detection and Quantification of ARGs and MGEs

In order to determine and quantify the presence of ARG, we extracted total DNA from the microbial communities present in the sediments and water from the sampled sites. The presence of ARGs was first screened by qPCR using the Microbial DNA Array for Antibiotic Resistance (Qiagen, Hilden, Germany) on PK and MC samples. Based on these results, a plasmid containing 14 ARGs was designed and constructed. The ARGs sequences inserted included resistance to tetracycline (*tetA, tetB*), β-lactam antibiotics (*bla*_OXA-2_*, bla*_IMP-2_*, bla*_IMP-5_*, bla*_IMP-12_*, acc-3*), methicillin (*mecA*), quinolones (*qnrB1*, *qnrB5*, and *qnrS1*), macrolide-lincosamide-streptogramin B (*msrA*), and sulfamethoxazole (*sul1* and *sul2*). The latter two were not included in the screening arrays but were previously reported in the area [26]. Also, the sequences of the bacteriophage crAssphage and *intl1* were inserted in the same plasmid. This plasmid was used to generate standard curves for all qPCR runs and samples analyzed.

The normalized abundance of these genes is shown in Figure 4. The *acc-3* genes were only detected in the Tagarete samples (UP2), both in sediments and water. The Class 1 Integron, together with *bla_OXA-2_* and *sul1* sequences, were detected in all the samples. The fecal contamination marker crAssphage was only detected in sediments and water from Uru Uru and its tributaries, except for UP1, in which it was only present in water. The UP2 site that receives wastewater discharges presented the highest crAssphage abundance. In contrast, crAssphage was not detected in PK and MC samples. In general, UP2 and UP3 samples were the ones with the highest abundance for the majority of the quantified genes, with *intl1*, *sul1*, *sul2,* and *bla*_OXA-2_ being the most abundant. Except for PK3, the hierarchical clustering analysis revealed a distinct gene abundance between the Milluni basin and Uru Uru sites.

The correlations of the normalized abundance among all detected genes were evaluated (Figure 5). *Intl1* was positively correlated with *sul1*, *sul2*, *bla*_OXA-2_, and *tetA* genes. Furthermore, these five genes also had a positive correlation among them, except *sul2* and *tetA*. The abundance of *tetA* presented an inverse correlation with *bla*_IMP-12_. Our analysis detected a strong correlation between crAssphage absolute abundance and *intl1*, *sul1*, *bla*_OXA-2_, and *tetA* but their Pearson coefficients were not significant.

### 3.3. Fecal Pollution and Antibiotic Resistance

To evaluate if fecal pollution contributes to the abundance of *intl1* and ARGs, a one-way ANOVA was performed to compare the abundance of *intl1* between samples with and without the presence of crAssphage. Samples in which crAssphage was detected presented a significantly higher abundance of *intl1* (*p* < 0.001). Moreover, the abundance of each ARG that positively correlated with *intl1* (i.e., *sul1*, *sul2*, *bla*_OXA-2_, and *tetA*) presented a statistically higher abundance in sampling sites with positive signal for crAssphage (Figure 6).

### 3.4. Relationship between ARGs and Environmental Variables

To evaluate the relationship between the abundance of ARGs, *intl1*, fecal pollution (crAssphage), and the environmental factors (metal levels and physicochemical parameters), a PCA was performed. ARGs and *intl1* abundance were reduced into the first two principal components (PCs) that explained 69.5% and 16.3% of the variation, respectively. A multiple Lm was performed in order to see if the ordination of the samples in function of the abundance of genes responded to any environmental variable along the PCs (Figure 7). The linear regression showed that the PC1 presented a positive linear relation with crAssphage, pH, and EC (Adj. R^2^ = 0.969; F = 50.52; *p* < 0.05), suggesting that fecal pollution, a neutral pH value, and high EC are the three conditions related with higher abundance of ARGs and *intl1*. No statistically significant relationship was found for the second PC. Remarkably, we could not find any significat association between metal levels and gene abundances.

## 4. Discussion

To our knowledge, the present study was the first to explore the relationship between ARGs, metal pollution, and wastewater discharges. In order to establish these relationships, we analyzed the abundance of different ARGs and the *intl1* gene in three water bodies. The PK site is a glacier lake that could be considered an intact ecological environment with very few anthropogenic activities around. Like PK, MC is also a glacier lake but is heavily impacted by mining discharges. Uru Uru, the third site, is a peri-urban lake with a long history of receiving both mining and wastewater discharges. Heavy metal levels were measured in sediment samples, and we also quantified the human fecal pollution marker crAssphage along with the abundance of different ARGs. Our results suggest that fecal contribution was the major driver of increased abundance of *intl1* and ARGs included in this study. These findings are in good agreement with a recent study that concluded ARGs abundance, in almost all cases, can be explained by the human fecal contributions in human-impacted environments and not by in situ selection proceses [18]. Despite the fact our results differ from other studies that favor metals as coselective agents of microbial antibiotic resistance [11,12,13,14,15], in situ coselection of antibiotic resistance driven by metals could still be playing an important role in these environments but would be masked by the effect of fecal discharges.

The absence of crAssphage suggests that PK is a pristine lake, while MC could be considered an extreme environment. MC samples presented very low pH, high EC, and elevated metal levels (Figure 2), which are characteristics associated with AMD-impacted sites [45]. These features were expected, as mining activities at large scales were performed in the Milluni area until 1990. Since then, only small cooperatives have operated in the valley [46]. These results support previous studies that indicate AMD discharges on surface water acidify and increase metal (As, Fe, Pb, Cd, Zn, Cu, Sn) levels [29]. Therefore, MC represents a harsh environment for microbial life, but remarkably, we were able to detect and quantify 4 out of 14 ARGs on this site. The Pata Khota Lake, on the other hand, presented low metal levels, almost neutral pH, and low EC, as expected for an intact ecological environment. The chemical composition of PK sediment is characteristic of natural lakes at high altitudes in the mountains [36]. Thus, the mineralogical composition explains the levels of metals found in this pristine site [29]. The Milluni Valley has little anthropogenic impact other than mining and camelid cattle raising for subsistence. Although it has been shown that the crAssphage used in our study can be found in water with animal fecal content [20], crAssphage could not be detected in Pata Khota and Milluni Chico (Figure 4). The latter could be explained by the extreme conditions that may influence the survival of crAssphage host in water. In Pata Khota, the absence of this fecal pollution marker could reflect either the absence of its host or the pristine character of this site.

In agreement with previous studies [30], Uru Uru Lake was characterized by high EC and alkaline pH. UP3 presented the highest levels of metals among all points, followed by MC sampling points. UP1, a point that receives water from an old open-pit gold mine transformed into an artificial meromictic lake, presented lower values of metals compared with MC points. This observation could also explain why UP1 EC values are more similar to PK, the site considered pristine. In the Tagarete channel (UP2), metal levels were very low and similar to those of PK Lake. UP2 receives wastewater discharges from Oruro city and disembogues in the Uru Uru Lake (UP3). Consistent with this fecal pollution input, all Uru Uru samples (except UP1 sediments) were positive for the crAssphage marker. Therefore, Uru Uru sampling sites were simultaneously impacted by both fecal pollution and mining discharges.

We analyzed the abundance of fourteen ARGs and the mobile genetic element *intl1.* Overall, seven ARGs were detected in our samples, and *intl1* was detected in all of them. The most abundant genes were *intl1*, *bla*_OXA-2_, *sul1,* and *sul2*. All these genes presented positive correlations among them (Figure 5). Commonly *intl1* can be found in the environment positively correlated with several ARGs [6] and its abundance is strongly correlated with the abundance of multi-drug resistant bacteria [47]. *intl1* and *s**ul1* are located together on MGEs and hence linked [28,48]. Previously, another study reported the presence of *sul1* and *sul2* in the Milluni valley, specifically in Pata Khota. When they analyzed the levels of sulfamethoxazole in water, the antibiotic was not detected [26] suggesting that the presence of these genes can occur naturally in bacteria residing in these aquatic environments, as has been found in other pristine sites [49,50,51]. Taking into consideration that PK and MC have little anthropogenic activity around, and that antibiotic levels are reported undetectable in one of our study sites [26] we assumed antibiotics do not play a major role in our analysis. Although we did not measure the antibiotic levels in the sampled sites, levels of erythromycin were reported under the limit of detection or in the order of ng/g of sediment for UP1 and UP2 nearby sites, respectively (Guzman-Otazo et al. In preparation). Even if other antibiotics could be present, their levels would be expected to be residual, given the dilution effect that they encounter in these large water bodies. Furthermore, recent evidence suggests that residual antibiotic concentrations play a secondary role regarding fecal contamination in ARGs abundance [18].

Some studies showed that metals such as Cu, Zn, Cd, and Ni could exert stronger selection pressure over environmental microbial communities favoring the selection of resistant bacteria, even more than antibiotics themselves [13,31]. Metal levels in our study were above the minimum coselective concentrations for antibiotic resistance reported in previous studies [52]. Even though metal levels in MC were higher than in PK, we found similar ARGs abundance in both lakes. In fact, they clustered together according to ARGs abundance (Figure 4) and were very close to each other in the PCA analysis (Figure 7). Samples collected in UP2 that drains sewage from Oruro city to UP3 presented the highest abundance of ARGs and *intl1*. Remarkably, there are significantly higher levels of all the measured metals in UP3. However, the ARGs abundance in both sites grouped in the hierarchical clustering analysis (Figure 4) These results suggest that, besides the known relationship between metals and ARGs, there are other parameters that might explain the variation in ARGs abundance in these sites.

Previous studies reported a positive correlation between ARGs and crAssphage [18,53,54]. Our hierarchical clustering analysis (Figure 4) revealed that Uru Uru samples were grouped together based on the presence of crAssphage and the highest levels of most of the ARGs. Although we found a positive but not statistically significant correlation between ARGs and crAssphage (Figure 5), samples with fecal pollution had a statistically significant higher abundance of *intl1*, *sul1*, *sul2*, *bla*_OXA-2_, and *tetA* than samples without fecal contamination (Figure 6). In addition, all these genes presented a strong positive correlation among them (Figure 5) which may indicate a possible common source. The influence of fecal pollution was clearly evident, as we found *acc-3* in the Tagarete channel (UP2). This gene is commonly associated with the human opportunistic pathogen *Hafnia alvei*, which is found in human and animal feces, water, soil and sewage [55,56]. Taken together, these results suggest that fecal discharges, as judged by the presence of crAssphage, might be an important source of AGRs in our samples. These results are in agreement with the use of crAssphage as a molecular marker able to track human fecal pollution in aquatic environments [20,57] and as a proxy to predict the presence of ARGs in wastewater impacted aquatic environments [53,54].

Our PCA results on *intl1* and ARGs abundance showed that PC1 (69.5% of the variation) had a strong linear relationship with *intl1*, *sul1*, *sul2*, *bla*_OXA-2_, and *tetA*. A multiparametric linear model revealed that the most important factors explaining the variation in this axis were EC, neutral pH, and crAssphage abundance. EC is an indicator of anthropogenic impact, as values increase with mining and sewage discharges [45,58]. It is well known that pH is considered the most critical factor influencing the microbial community composition in soils [59,60]. Furthermore, microbial community composition is the most important factor determining the resistome in soils at continental levels [61]. It is important to note that pH values were only measured in the water of each sampling site. Although overlying waters do not necessarily correlate with the pH of sediments in acidified environments [62], previous studies on the Milluni basin [29] reported similar sediment pH values. On the other hand, although this analysis could not find any relationship between metal levels and the abundance of these genes, it is known that toxicity and bioavailability of heavy metals depends on its solubility, which increases at low pH values [52]. Long-term heavy metal pollution at high concentrations reduces microbial diversity and biomass [63]. Therefore, it is likely that heavy metals, pH, and EC indirectly condition the resistome in mining and wastewater impacted environments.

Overall, our results suggest that fecal pollution is the major driver of ARGs and *intl1* abundance in aquatic environments impacted by both mining and wastewater discharges. pH and EC potentially influence the resistome in an indirect manner. Metal levels in all our samples were above minimum coselection concentrations. Thus, we cannot rule out in situ coselection processes. Nonetheless, our results strongly suggest that sewage introduced bacteria is enough to mask selection effects in the local resistome. Further analysis, especially metagenomic approaches, are needed to clarify whether the abundance of ARGs, including those not detected in our analysis, is related to fecal pollution or metal contamination in waterbodies impacted by both, mining activities and wastewater.

## Figures and Tables

**Figure 1 microorganisms-08-01122-f001:**
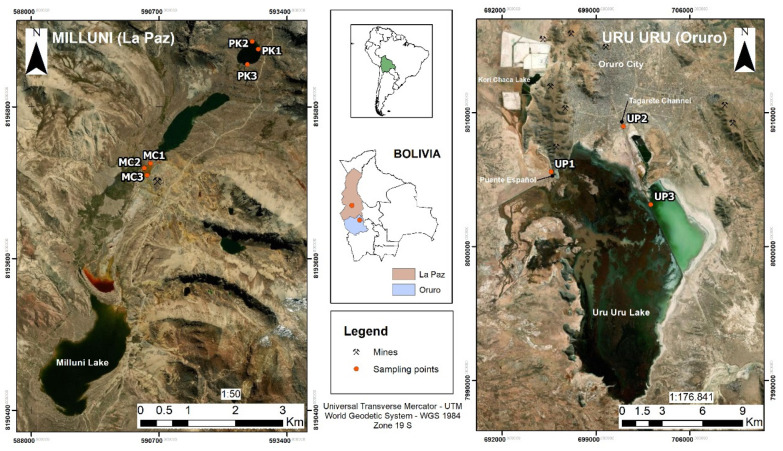
Study area. The left panel shows the sampling sites in the Milluni Valley. Three samples were collected from both lakes, the pristine Pata Khota (PK) and Milluni Chico (MC). MC is located directly downstream of the Mine “*La Fabulosa*.” The right panel shows the sampling sites in the Uru Uru Lake, in the department of Oruro: UP1 is located at the channel that discharges the water of the meromictic lake Kori Chaca an old open-pit gold mine, UP2 is located at the Tagarete channel that carries untreated wastewater discharges from Oruro city, and UP3 is located in the northeast part of Uru Uru Lake, and receives the discharges of both mining residues and wastewater.

**Figure 2 microorganisms-08-01122-f002:**
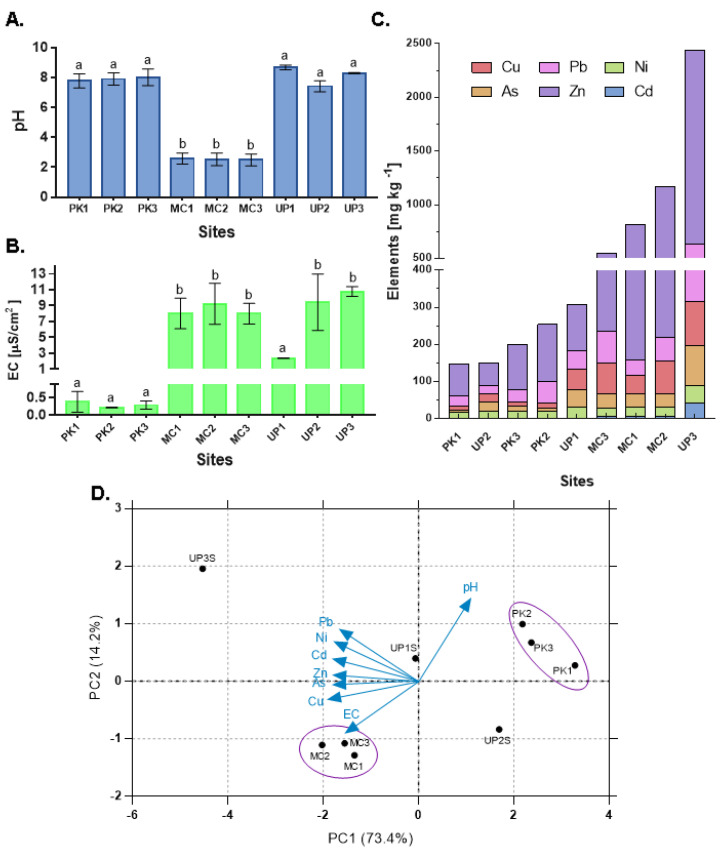
Physicochemical parameters and metal levels. The mean values are shown in bars, and standard deviation values are presented as the error-bars. Different letters represent statistically significant differences (*p* < 0.05), calculated by analysis of variance (ANOVA). (**A**) pH and (**B**) EC of water. (**C**) The levels of metals (mg Kg^−1^) measured from sediment samples. The data is shown in bars with cumulative values. (**D**) PCA of physicochemical parameters and metal concentrations of sampling sites.

**Figure 3 microorganisms-08-01122-f003:**
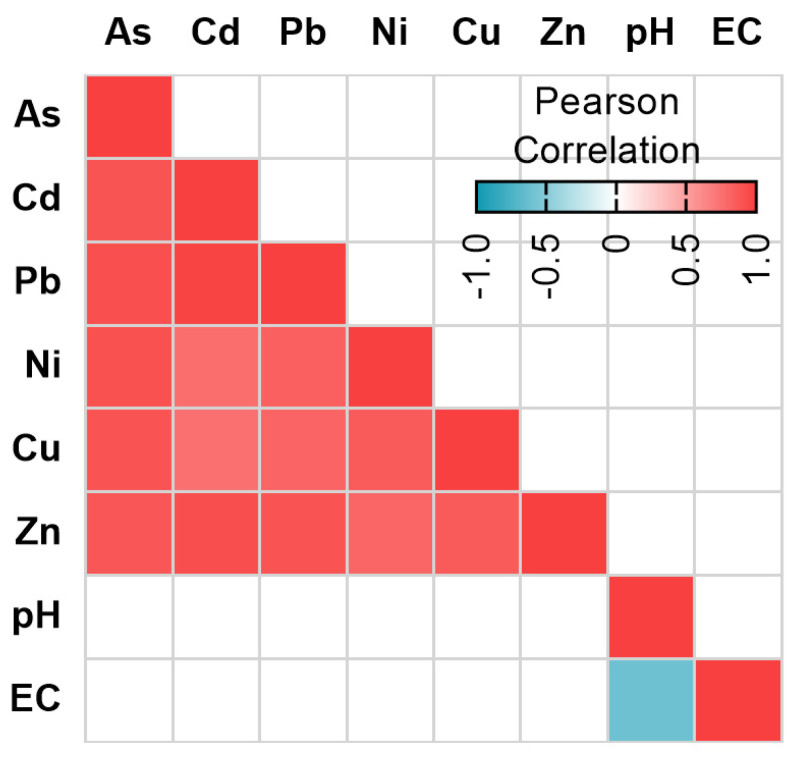
Correlation among metals and physicochemical parameters. The R correlation coefficient is represented in colors, as indicated in the legend. Only significant correlations (*p* < 0.05) were included.

**Figure 4 microorganisms-08-01122-f004:**
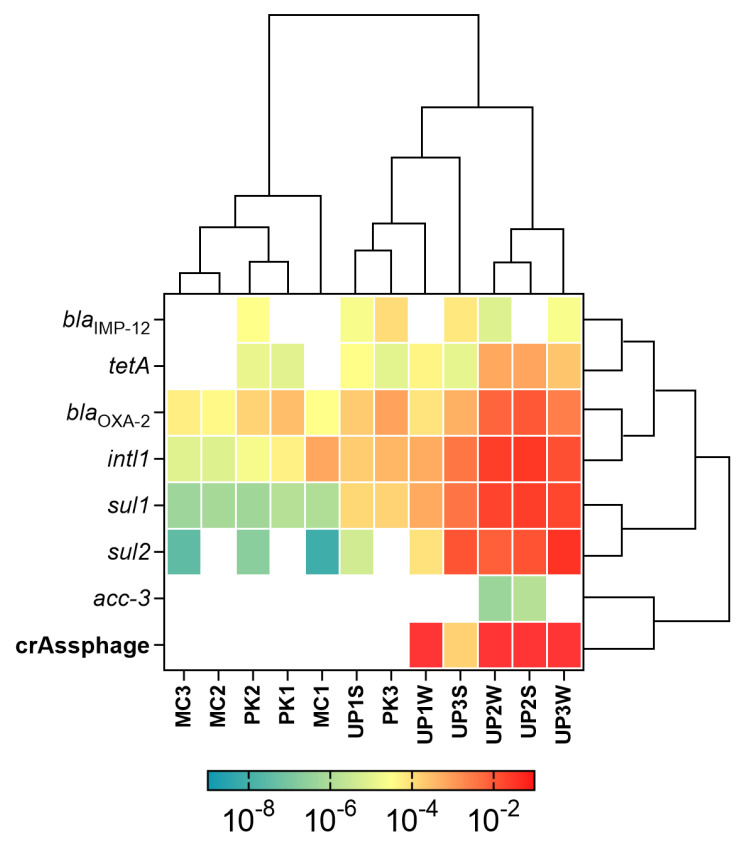
Normalized abundance of ARGs, *intl1*, and crAssphage detected on sediments and water. The abundance values of ARGs and *intl1* were normalized to 16S rRNA gene abundance, and the absolute abundance of crAssphage was included. The data were transformed using Log_(10)_ and represented in a heatmap where reddish coloration symbolizes a higher abundance. Rows and columns were ordered by similarity with hierarchical clustering. S: Sediments; W: Water.

**Figure 5 microorganisms-08-01122-f005:**
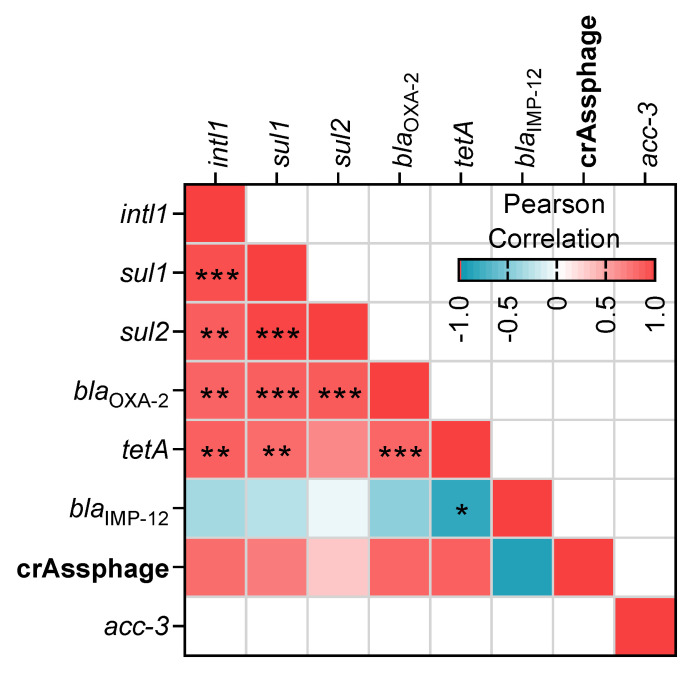
Correlation among the detected genes. R correlation coefficient is represented as a heat map of colors. Significative correlations are depicted as follows: * *p* < 0.05, ** *p* < 0.01, *** *p* < 0.001.

**Figure 6 microorganisms-08-01122-f006:**
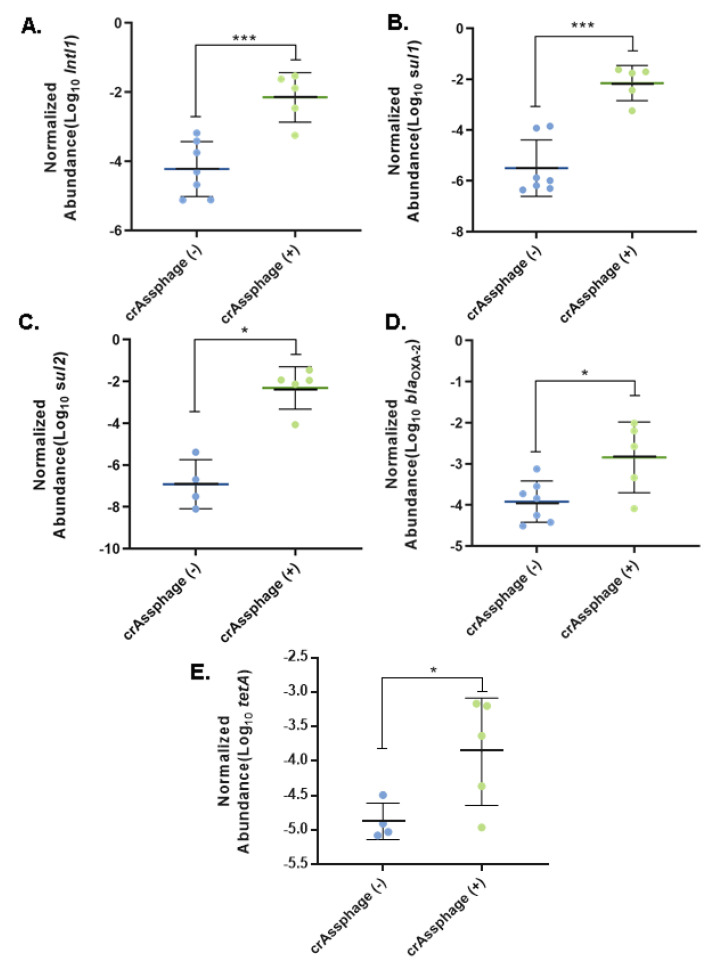
Abundance of ARGs and *intl1* in function to the presence of crAssphage. The abundance of (**A**) *intl1*, (**B**) *sul1*, (**C**) *sul2*, (**D**) *bla*_OXA-2_, and (**E**) *tetA* between the samples with and without detected crAssphage were compared using ANOVA. * *p* < 0.05, ** *p* < 0.01, *** *p* < 0.001.

**Figure 7 microorganisms-08-01122-f007:**
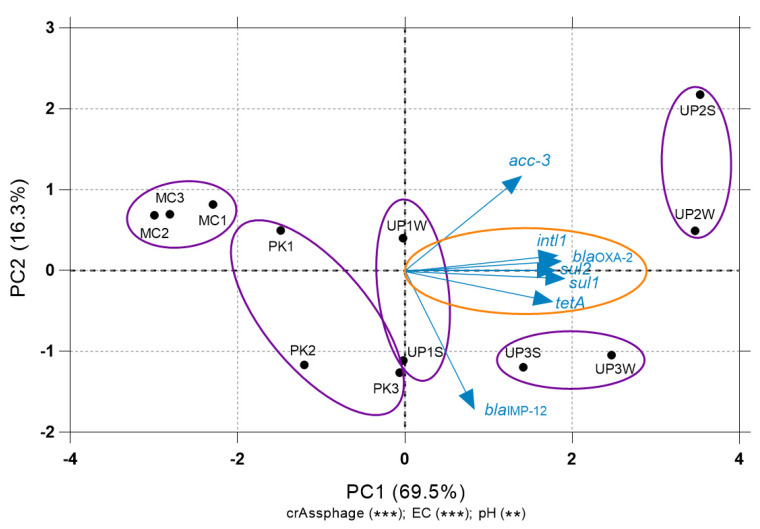
Relationship between environmental variables, *intl1*, and ARGs abundance. The PCA was performed with the data of normalized gene abundance per site of ARGs and *intl1*. A multiple Lm of the PCs with environmental variables was performed. The numbers in parenthesis represent the percentage of variation explained by the axis, and the parameters that are significantly related to the axis are expressed with: * *p* < 0.05, ** *p* < 0.001, and ****p* < 0.0001. Related sampling points are indicated within purple circles. The abundance of *intl1* and the ARGs are represented as blue arrows. The direction of the arrow indicates an increasing abundance of the genes. The angle of the arrows with respect to the axis represents the linear relation of the abundance with the PC, and the orange circle shows the most important variables (*intl1*, *sul1*, *sul2*, *bla*_OXA-2_, *tetA*). Along the PC1, the abundance of *intl1* and ARGs increased toward the right. S: Sediments; W:Water.

**Table 1 microorganisms-08-01122-t001:** Primers used for quantitative polymerse chain reaction (qPCR) experiments.

Gene	Forward (5′ → 3′)	Reverse (5′ → 3′)	Size (bp)	Ref.
*sul1*	GGATTTTTCTTGAGCCCCGC	CACCGAGACCAATAGCGGAA	99	This study
*sul2*	TCATCTGCCAAACTCGTCGT	CAAAGAACGCCGCAATGTGA	103	This study
*bla*IMP*-5*	CTTGGTTTGTGGAACGCGG	TAAGCCACTCTATTCCGCCC	87	This study
*bla*IMP-2	GAGCGCGTTTGCCTGATTTA	AGAAACAACACCCCAACCGT	95	This study
*bla*IMP-12	TGAAGAGGTTAGCGGTTGGG	CGCCCTACAAACCAAGCAAC	132	This study
*bla*OXA-2	GGTAGGATGGGTTGAGTGGC	ATAGAGCGAAGGATTGCCCG	120	This study
*acc-3*	GTTGCTACGCCGATTGTTCC	GCGATGTAGGCACCAAAACC	92	This study
*tetA*	TCAATTTCCTGACGGGCTG	GAAGCGAGCGGGTTGAGAG	91	[42]
*tetB*	AGTGCGCTTTGGATGCTGTA	GCTGAGGTGGTATCGGCAAT	98	This study
*msrA*	CTGCTAACACAAGTACGATTCCAAAT	TCAAGTAAAGTTGTCTTACCTACACCATT	89	[43]
*mecA*	GGTTACGGACAAGGTGAAATACTGAT	TGTCTTTTAATAAGTGAGGTGCGTTAATA	106	[43]
*QnrB-1*	GCGGCACTGAATTTATCGGC	GGCATCTTTCAGCATCGCAC	86	This study
*QnrB-5*	CGGGGTGTTGATTTACAAGGC	GCCAATAATCGCGATGCCAA	84	This study
*intl1*	CAGCACCTTGCCGTAGAAGA	GAGGCATTTCTGTCCTGGCT	99	This study
crAss- phage	AGGAGAAAGTGAACGTGGAAACA	AACGAGCACCAATTTTAAGCTTTA	78	Modified from [20]
16S rRNA	CCTACGGGAGGCAGCAG	TTACCGCGGCTGCTGGCAC	192	[44]

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
