# Peer review of "Fecal Pollution Drives Antibiotic Resistance and Class 1 Integron Abundance in Aquatic Environments of the Bolivian Andes Impacted by Mining and Wastewater"

_microorganisms, 2020, doi:10.3390/microorganisms8081122_

Round 1

Reviewer 1 Report

The manuscript entitled “Antibiotic resistance genes and class 1 integron:  Evidence of fecal pollution as a major driver for their abundance in water and sediments impacted by metal contamination and wastewater in the Andean region of Bolivia” by Agramont and co-authors in which the authors have assessed the higher prevalence of antibiotic resistance genes and class 1 integrons in the aquatic environment and they concluded that it was due to anthropogenic activity and alteration in physiological conditions. The authors have described all the experimentation in detail and the results are well described but the results need to discuss in detail in corroboration with references. This is an interesting article for the Microorganisms journal. However, there are a few concerns with this manuscript. The authors need to take care of scientific language and try to avoid redundancy. The authors need to specify why they were targeting only class1 integron, why not other integron classes.  

Specific comments:

The title of the manuscript is too long.

The abstract is not clear; it is suggested to rewrite it.

The authors have described briefly the outcome of this study in the introduction section (Page 3; Line: 85-91). Kindly delete that.

The figure legends are much elaborated. The authors need to describe it briefly.

If the authors have abbreviated something in the first stance then abbreviations should be consistently used throughout the manuscript.

Reviewer 2 Report

This study tries to analyze the influence of metal pollution, and human fecal discharges on the abundance of different ARGs and the Intl1, in water and sediments samples from a pristine, metal-polluted and wastewater-mining contaminated lakes to determine the contributions of metals and fecal discharges in the abundance of ARGs. The authors suggest that fecal pollution is the major driver of higher ARGs and intl1 in wastewater and mining contaminated environments. Actually, several recent reports have indicated the positive correlations of ARGs and crAssphage, and three reports have been cited in current MS. For example, a recent study “Fecal pollution can explain antibiotic resistance gene abundances in anthropogenically impacted environments (Nature Communications volume 10, Article number: 80 (2019)” demonstrates that the presence of antibiotic resistance genes can largely be explained by fecal pollution, with no clear signs of selection in the environment, with the exception of environments polluted by very high levels of antibiotics from manufacturing, where selection is evident. This important reference is also cited in current study. Thus, it seems to me the current study is actually a confirmatory study but it still provides supporting data compared to previous studies. In general, the findings in this study could support the suggestions claimed by the authors but some issues need to be clarified. For example, the size of sample may be too lees to explain the relationship between the human pollutions and spreading of ARGs?

1. The authors need to explain more about levels of metals as selective agents of microbial antibiotic resistance to could strengthen their findings.

As already described in this manuscript, some previous reports have found the relationship between levels of metals and antibiotic resistance. And, in this MS, samples of UP3 and UP2 also contains relatively higher concentration of metals. Possibly the metals in the two sites could still play roles to affect the presences of antibiotic resistance genes. Indeed, the authors also indicated that “Therefore, Uru Uru sampling sites are simultaneously impacted by both fecal pollution and mining discharges (line 412-415)”, and also in line 434-443, the authors already explained that “These results suggest that other parameters different from metal levels are explaining the variation in ARGs abundance”. However, the authors should explain more about the metal effects and crAssphage on antibiotic resistance genes spreading. Firstly, is the co-selection effects of metal and antibiotic resistance could be explained by the levels of metal concentration? Second, what’s the criteria for those metal pollutions in water? Finally, howe about the single or the combination effects of different metal irons on antibiotic resistance?

2. In line 378-385. The authors indicate that “This conclusion offers a different scenario from what has been suggested in other studies that favor metals as selective agents of microbial antibiotic resistance [12-15], through co-selection processes in the environment [11]. However, it is important to note that fecal pollution was not considered in any of these studies”. In fact, quite a few ARGs (although relatively low abundance) were all detected in samples of MC1, MC2, MC3, PK1, PK2, and UP1, and these samples were not detected with crAssphage. Thus, it seems to me, the metals could be still selective agents for microbial antibiotic resistance in some extent.

3. In line 444-457: there are only 9 samples points in this MS, and the crAssphage are only positive in three samples, and this study observed that all ARGs genes presented a strong positive correlation among them, but no individual correlation between the abundance of crAssphage and the other genes were found. The authors explained that the above findings could be explained by the low number of crAssphage positive samples (only five). In fact, among those five samples, UP1 and UP3 contain relatively higher level of metal concentrations in nature, and in figure 4, UP1F (this could be UP1S or UP1W?), UP3S, UP2W, UP2S, and UP3W are actually originated form the same three samples. Thus, it seems to me, the size of sample is too lees to explain the relationship between the human pollution and spreading of ARGs?

4. In figure 4, what’s the “UP1F”? this could be UP1S or UP1W?

Round 2

Reviewer 1 Report

The authors have addressed all the comments and substantially improved the manuscript. This manuscript can be accepted for publication in Microorganisms.    

Reviewer 2 Report

The authors have satisfied my concerns; the manuscript is now acceptable for publication